

# The validity and reliability of the Test of Memory Strategies among Italian healthy adults

Maria Grazia Vaccaro[1,2,*], Marco Tullio Liuzza[1,*],
Massimiliano Pastore[3], Nuria Paúl[4], Raquel Yubero[5],
Andrea Quattrone[1], Gabriella Antonucci[6,7], Antonio Gambardella[8] and
Fernando Maestú[9]

[1] Department of Medical and Surgical Sciences, Magna Græcia University of Catanzaro, Catanzaro, Calabria, Italia
[2] Neuroscience Center, "Magna Græcia" University, Catanzaro, Italy, Catanzaro, Calabria, Italia
[3] Department of Developmental and Social Psychology, Padova University, Padova, Veneto, Italy
[4] Department of Experimental Psychology, Universidad Complutense de Madrid, Madrid, Spain
[5] Department of Neurology, Quirón Pozuelo Hospital, 28223, Pozuelo de Alarcón, Madrid, Spain
[6] Department of Psychology, University of Roma "La Sapienza", Roma, Lazio, Italy
[7] Fondazione Santa Lucia, IRCCS, Roma, Lazio, Italia
[8] Department of Medical and Surgical Sciences, "Magna Græcia" University of Catanzaro, Catanzaro, Calabria, Italia
[9] Networking Research Center on Bioengineering, Biomaterials and Nanomedicine (CIBER-BBN), Universidad Complutense de Madrid, Madrid, Spain
* These authors contributed equally to this work.

Corresponding author
Marco Tullio Liuzza, liuzza@unicz.it

## ABSTRACT

**Background:** Previous literature has shown that executive functions (EF) are related to performance in memory (M) tasks. Nevertheless, there is a shortage of psychometric tests that examine these two constructs simultaneously. The Test of Memory Strategies (TMS; previously validated in Spain and Portugal) could be a useful verbal learning task that evaluates these two constructs at once. In this study, we aimed to evaluate the psychometric properties of the TMS in an Italian adult sample.

**Method:** One hundred twenty-one healthy volunteers (74 F, Mean age = 45.9 years old, SD = 20.4) who underwent a neuropsychological examination participated in this study. We conducted a Confirmatory factor analysis (CFA) to evaluate the structural validity of the TMS. We conducted a latent variable analysis to examine convergent and discriminant validity of the TMS sub-scale scores reflecting executive functions and memory. We also examined the TMS reliability in terms of internal consistency through the McDonald's omega.

**Results:** The CFA confirmed the expectation that the TMS-1 and TMS-2 subtests reflect a factor and that the TMS-3, TMS-4, and TMS-5 subtests reflect a different factor. This result is in line with the prediction that TMS-1 and TMS-2 require the use of executive functions and memory simultaneously, and therefore we called this factor executive functions (EF); whereas the TMS-3, TMS-4, and TMS-5 subtests require less involvement of executive functions, thus reflecting a construct that we named memory (M). The TMS subtests for EF and M showed convergent validity with the test scores using a traditional neuropsychological battery, assessing memory and executive functions separately. Finally, the reliability of the subtests was good.

**Conclusions:** These preliminary findings suggest that TMS is a valid and reliable scale to simultaneously assess M and EF while among Italian healthy adults.

# INTRODUCTION

Previous research has shown that executive functions are related to performance in episodic memory (M) tasks (*Brooks, Weaver & Scialfa, 2006*). It is always hard to interpret whether a failure in a memory test is mainly due to a primary memory deficit or to an impairment of executive functions (EF). Previous literature has observed that the EF has a vital role in the performance of M tasks (*Brooks, Weaver & Scialfa, 2006*; *Li et al., 2016*). *Craik et al. (2018)* conducted two studies investigating these two important aspects of memory problems in older adults who have difficulty retrieving recent episodic events and an often-transient inability to retrieve names and other well-known facts from semantic memory. In fact, one of the questions they asked was whether these age-related difficulties reflected a common cause: a recovery problem related to inefficient EF. From their studies it emerges that no task is a pure measure of the theoretical constructs of EF or of episodic and semantic memory. Their studies showed that individual differences in EF in older adults correlate with the effectiveness of recovery in both episodic and semantic memory, but also that these relationships depend on the specific tests chosen to represent both EF and the memory recovery (*Craik et al., 2018*). Therefore, executive function and memory deficits occur in most neurodegenerative and psychiatric disorders, and many other diseases (*Arevalo-Rodriguez et al., 2015*; *Khan et al., 2014*; *Litvan et al., 1994*). A mediation analysis revealed that the EF network had an indirect positive effect on episodic memory performance in the amnestic mild cognitive impairment patients (*Yuan et al., 2016*). These findings provide new insights into the neural mechanisms underlying the interaction between impaired EF and M deficits in amnestic mild cognitive impairment patients and suggest that the EF network may mediate episodic memory performance in patients with mild cognitive impairment (*Yuan et al., 2016*).

Other studies, however, such as *Moro et al. (2015)* utilized a training program to teach specific strategies and metacognitive skills to enable patients to perform attention and executive tasks. When training was performed on attention and executive functions, the results showed generalized improvements also to memory (*Moro et al., 2015*). Despite brain imaging and neuropsychology findings, there is a shortage of psychometric tests that examine the interaction between EF and M constructs. Until recently, there were not many neuropsychological tests to evaluate the interaction between EF and M functions. The assessment of this interaction would be helpful to understand whether a bad performance on an episodic memory test is due to a primary impairment on executive functions or memory (*Bäckman, 2008*; *Saenz et al., 2015*; *Stramaccia et al., 2018*; *Yubero et al., 2011*). With few exceptions, such as the Rey Auditory Verbal Learning Test (RAVLT,

Carlesimo, 2014), California Verbal Learning Test (CVLT-II, Woods et al., 2006), short story recall (Carlesimo et al., 2002), and Wechsler Memory Scale (Dumont et al., 2014), classical neuropsychology views have promoted the use of separate tests (Goretti et al., 2014; Mattioli et al., 2014) to assess M and EF in different conditions and diseases (Lis et al., 2008; Vaccaro et al., 2018). However, none of them assess the interaction between EF and memory in a parametric way. To fill this gap, Yubero et al. (2011) built the Test of Memory Strategies (TMS) to evaluate the impact of EF on performing a memory task. This tool allows us to test whether a deficit found in a memory task could be associated with a primary memory problem or whether that deficit could be due to an executive problem affecting memory performance. TMS presents a type of verbal learning task in which, through consecutive stages, the need to enact internal memory strategies should be progressively reduced. Yubero et al. (2011) in their study hypothesized that patients with severe EF damage would score worse when more internal memorization strategies were needed to complete a task; conversely, participants with a greater deficit in memory functions would have scored worse when memory involvement was predominant. While participants with impaired EF and memory would have scored very low at any stage of the TMS (Yubero et al., 2011).

In detail, Yubero and colleagues in their study applied TMS to four groups of elderly patients with varying levels of cognitive impairment. Under conditions with low material organization, the multidomain groups of mild cognitive impairment and vascular cognitive impairment showed greater impairment of executive function. However, as the material was progressively organized, the multidomain mild cognitive impairment and vascular cognitive impairment groups improved their performance. The study results confirm executive functions appear to influence performance in memory tasks (Yubero et al., 2011).

In 2018, Fernandes et al. (2018) the authors translated and validated the TMS for use on the Portuguese population. The focus of Fernandes et al. (2018) was to analyze the performance of TMS in healthy individuals of different ages, to study the test's ability to discriminate between different age groups and education levels. In their study, Fernandes and colleagues hypothesized that the growing external organization of TMS activities would affect the response of the older sample in a negative way.

Fernandes et al. (2018), like Yubero et al. (2011), highlighted the importance of the external organization of activities in the different phases of the TMS. In addition, we expected groups with higher formal education levels to perform better in TMS than those with fewer years of formal schooling. Finally, they showed adequate psychometric properties of TMS (Fernandes et al., 2018). The results showed the reliability of TMS scores based on internal consistency analysis. Factor analysis of the TMS scores revealed the test produced two factors, one capturing executive function and the other capturing memory. Correlations with classical neuropsychological tests supported the convergent and discriminant validity of the TMS scores. The older groups had greater difficulty in creating and mobilizing strategic memory than the younger group after controlling for the influence of education, although both groups experienced performance improvements in the five TMS sub-tests. Fernandes and colleagues, stated that the results of their study

suggest that TMS is an adequate measure for assessing memory and EF while applying it to a Portuguese sample (*Fernandes et al., 2018*).

Some tests investigate the executive functions and memory separately (such as Rey Auditory Verbal Test; *etc.*), but this does not allow to verify if during execution the subject fails because of a slowdown in memory or planning or problem-solving strategies. Instead, the TMS is structured to capture which phase of the test the subject increased the number of errors. A failure in the initial condition would but a clear improvement after diminished executive functions would signify an executive deficit more than a memory problem. If the participant does not improve across time, then a primary memory problem can be noticed.

In the adult and childhood clinical population, it is difficult to understand whether there is a memory or executive function impairment or both just from the interpretation of traditional neuropsychological tests. This study aimed to improve and increase the neuropsychological assessment tools available in Italy to cognitive diagnosis using the TMS, a neuropsychological test already validated in Spanish and Portuguese (*Yubero et al., 2011*; *Fernandes et al., 2018*). There is no neuropsychological test for evaluating the interaction between memory skills and executive functions simultaneously in the Italian context. This preliminary study on the Italian population could fill the gap in the neuropsychological test resources for improving diagnosis. The ability to generate memory strategies is a key factor in the performance of episodic memory tests. *Fernandes et al. (2018)* argue that there is evidence about the inefficient use of memory strategies in older adults. However, it is not clear whether a worse performance on a memory test in older people might be attributed to an inability to mobilize cognitive strategies or to an episodic memory deficit (*Fernandes et al., 2018*; *Yubero et al., 2011*). We contend that in the Italian literature there are no validated tests adapted for an Italian sample that measures memory and executive functions simultaneously and allows to detect, in specific conditions, when, for example, there is primary damage of memory or executive functions. This study tried to address the question by validating the TMS—which parametrically reduces the need for executive functions on memory tests—on an Italian sample. The main aim is to have a version of the TMS available for the Italian population to help achieve better cognitive diagnosis.

# MATERIALS AND METHODS

## Participants

We enrolled a sample of 121 participants (74 F) between 18 and 89 years (Mage = 45.90 years old, SD = 20.40; years of schooling = 13.10 ± 3.97); with an average MMSE score of 29.10 ± 1.59. Participants were divided into four age groups, 18–30 (G1), 31–50 (G2); 51–60 (G3), 61–89 (G4). The inclusion criteria were that participants were healthy, no cognitive impairment interfering with their daily activities; no neurological and psychiatric disorders; no systemic disease (*i.e.*, diabetes), or drug addiction; and no illiteracy. The Mini Mental State Examination (MMSE) was used to select patients without cognitive impairment but the scores of some test subscales were also used for statistical analyzes (MMSE, *Folstein, Folstein & McHugh, 1975*; *Magni et al., 1996*; *Mazzi et al., 2019*); the cutoff point of 24 was used for all participants selection. The participants provided written

informed consent to take part in the validation and the procedures were carried out in accordance with the principles of the Declaration of Helsinki. All procedures were in accordance with the ethical standards of the Italian National Committee (AIP—Associazione Italiana di Psicologia) and under the ethical standards of the CEIC Hospital Clínico San Carlos (Madrid—Spain) with number C.I. 18/422-E_BS. We got informed consent from all individual participants included in the study. In general, the data was collected with previous work on TMS as a guide, such as *Fernandes et al. (2018)*.

Neuropsychological Assessment We evaluated the general cognitive functions of all participants through a battery of standardized neuropsychological tests validated on the Italian population. The following tests for the executive functions category were administered: MMSE subscale about "attention and calculation"; verbal fluency (VF), lexical stock, ability to access the lexicon, and cognitive flexibility (*Patterson, 2018*; *Bianchi & Dai Prà, 2008*); semantic fluency (SM—*Bianchi & Dai Prà, 2008*) and the higher the score obtained by the subjects, the better the performance; Stroop Color and Word Test (Stroop) (*Scarpina & Tagini, 2017*) evaluated attention with an interference procedure (times in seconds), and the higher the score obtained by the subjects, the worse the performance; the Modified Wisconsin Card Sorting Test (MWCST) was a modification of the original Wisconsin Card Sorting Test (*Laiacona et al., 2000*; *Vaccaro et al., 2018*) was used to measure executive functions, and the higher the score obtained by the subjects, the better the performance. The following tests for the memory category were administered: the "Registration" and "Recording" subscale MMSE the Digit Span (DS) (*Orsini et al., 1987*) subtest from the Wechsler Adults Intelligence Scale (WAIS-IV) (*Orsini & Pezzuti, 2013*, *2015*) was used to assess the short-term verbal memory and working memory strategies, and the higher the score obtained by the subjects, the better the performance; Wechsler Memory Scale (WMS) was used to assess immediate verbal learning and delayed memory (*Reynolds & Powel, 1988*), and the higher the score obtained by the subjects, the better the performance. The traditional neuropsychological tests were used to analyze the construct validity of the Test of Memory Strategies (TMS) (*Yubero et al., 2011*). However, we removed the "Recalling subscale" of the MMSE from the analyses because all the participants but one had a ceiling performance (three out of three), and therefore there was insufficient variability in that variable.

## Test of memory strategies

The description of the TMS tool follows the detailed description which has already been reported by *Fernandes et al. (2018)* and as described below. The TMS consists of five word lists (see the Supplementary Materials) to be learned by the subject (TMS-1, TMS-2, TMS-3, TMS-4, and TMS-5), and each individual list consists of 10 words. The aim of the TMS is to evaluate the influence of EF and M in cognitive test performance. The test consists of different experimental conditions in which the need for EF is progressively diminished, with the probability of observing an increase in M performance in subjects with primary EF deficits. On the contrary, it is not expected to observe an increase in EF performance in subjects with primary M deficits. The five lists of the TMS are as follows (*Fernandes et al., 2018*; *Yubero et al., 2011*): TMS-1: an incidental learning task consisting of 10 words with

no semantic and/or phonetic relationship between them; TMS-2: same condition as TMS-1.; TMS-3: a task with ten words belonging to two semantic categories—trees and furniture. The words are presented randomly in the TMS-2 and TMS-3 conditions, the need for working memory should be most demanded; in TMS-4, however, the words are always organized into two semantic categories, but unlike in TMS-3, the words are presented in an ordered manner for the categories transport and tools (from work). A reduction in memory strategies should occur because the material is organized externally into two consecutive semantic categories. Finally, in TMS-5 the words are organized as in TMS-4 however, the instructions are different. It is explicitly reported to the participant that there are two distinct semantic categories but without specifying which one they are (sports and vegetables). Again, less involvement of internal cognitive strategies should be required due to the external organization of the material. In details, the data were collected as previously described by *Fernandes et al. (2018)*.

The instructions given to the participants were the same for each word list from List 1 to List 4 and different in the TMS List 5 instructions (*Fernandes et al., 2018*; *Yubero et al., 2011*). In each TMS list, words are read at the rate of one word per second. Each recalled word receives a point, with the total score ranging from 0 to 10 for each list and from 0 to 50 for the total scale (sum of the five-word lists) (*Fernandes et al., 2018*; *Yubero et al., 2011*). The word lists (from TMS-1 to TMS-5) were read to the participants one immediately after the other. There is no latency or delay between one word list and another. In the present study, the original TMS Spanish version was translated into the Italian version in accordance with the international guidelines for translation and cross-cultural adaptation (*Guillemin, Bombardier & Beaton, 1993*). This form was compared with the original one, and subsequently, the original author approved the translated version of the Italian TMS. The original Spanish version was forward-translated by two independent translators—a Spanish expert (M.R.) with DELE exam of Cervantes Institute and a psychologist fluent in Spanish with experience in research and DELE exam of Cervantes Institute, and their translation agreed with a final Italian version. The Spanish back-translation was compared to the original version to detect any misinterpretation and ambiguity; the two versions were found to be reasonably similar. Furthermore, the Italian translation was compared to the original version to ensure conceptual equivalence and improve understandability. We chose to keep these same words from the original list, such as the one used to describe in the work of *Yubero et al. (2011)* because they were similar in both languages and their frequency of occurrence was similar in the common Italian language. The evaluation was conducted in a single session lasting approximately 90 min.

## Statistical analysis

The statistical analyses were performed using free and open statistical software, Jamovi (version 1.6.15, 2020) and through the R statistical programming environment (*R Core Team, 2022*). Descriptive statistics of TMS and neuropsychological test were calculated for the whole sample, including frequencies, means, skewness, kurtosis, and minimum and maximum score obtained for each list by the participants.

Correlations between age and education and the different scores obtained by the subjects in each single list of the TMS (TMS List 1 tot; TMS List 2 tot; TMS List 3 tot; TMS List 4 tot; TMS List 5 tot), and with the total score obtained by adding the scores of each single TMS list (TMS tot Lists) were calculated. Correlations between the two subscales (M and EF) and classical neuropsychological tests were calculated. Finally, Bonferroni correction to the correlation analysis was applied for each correlation matrix by multiplying the $p$ value by the $k = k(k − 1)/2$ pairwise correlations.

We conducted an ANOVA on the EF and M subscales as dependent variables for age groups. Correct *post hoc* comparisons were then performed for Tukey correction.

We tested whether the TMS list total scores were multi-normally distributed through the *Mardia (1970)* for multivariate skewness and Kurtosis, followed up by the Shapiro-Wilk tests for univariate normality on the single variables. Mardia's test showed that the multivariate skewness assumption is not tenable (*test statistic* = 68.03, $p < 0.001$), whereas the multivariate normality kurtosis holds (*test statistic* = 0.18, $p = 0.85$). The follow-up analyses showed that univariate normality was not met for any of the variables ($Ws > 0.95$, $ps < 0.01$).

We conducted a set of confirmatory factor analysis (CFA) performed using maximum likelihood robust (MLR) estimator. We compared different models to identify the model that best fits our data using the Bayesian Information Criterion (*Schwarz, 1978*), which penalizes overfitting. These analyses were conducted using the *lavaan* package in R (*Rosseel, 2012*).

We hypothesized that TMS lists reflected two-dimensional structure with two latent variables, that we have named as EF, and M. We have considered the total score of each single five lists indicators, and we tested whether the hypothesized model where the EF factor was reflected by the first two TMS list scores (TMS-1; TMS-2), and the M factor was reflected by the TMS list scores number 3, 4 and 5 performed better than the other three alternative models. The first alternative model was a tridimensional model (Panel C) with a third factor (EFM) reflected by the TMS 3 score. From this model emerges the idea that TMS reflects a sort of further latent variable where EF and M are mixed. The second alternative model was a unidimensional model (Panel D). This represents the most parsimonious model and reflects the idea that the TMS lists do not reflect EF and M differently. Finally, the third model was an alternative two-dimensional model (Panel B) in which list 3 (TMS-3) was considered a variable reflecting the factor EF instead of M. Finally, a fifth alternative model was calculated which included the first four lists (TMS-1; -2; -3; 4) representative of the factor EF and TMS-5 of the M factor. This model was chosen because although there is a semantic relationship between two categories of words, the material is not organized as in the TMS-4 list. Indeed, the fact that the words belonging to the same category are not ordered could require the subject to use a strategy, and therefore the intervention of executive functions rather than memory.

Goodness of fit was assessed through standardized root mean square residual (SRMR), root mean square error of approximation (RMSEA), Tucker–Lewis fit index (TLI), and comparative fit index (CFI, *Bentler, 1990*). Model acceptability was evaluated through the
**Table 1 Descriptive statistics about subscales and total scores of TMS.**

|  | TMS List1 total | TMS List1 interferences | TMS List 2 total | TMS List2 interferences | TMS List3 total | TMS List3 interferences | TMS List4 total | TMS List4 interferences | TMS List5 total | TMS List5 interferences | TMS total list |
|---|---|---|---|---|---|---|---|---|---|---|---|
| N | 121 | 121 | 121 | 121 | 121 | 121 | 121 | 121 | 121 | 121 | 121 |
| Mean | 3.61 | 0.298 | 4.40 | 0.388 | 4.84 | 0.339 | 5.73 | 0.504 | 5.21 | 0.645 | 23.8 |
| Standard deviation | 1.42 | 0.542 | 1.51 | 0.583 | 1.95 | 0.571 | 2.20 | 0.732 | 1.99 | 0.773 | 7.34 |
| Minimum | 0 | 0 | 1 | 0 | 1 | 0 | 0 | 0 | 1 | 0 | 6 |
| Maximum | 7 | 3 | 9 | 3 | 9 | 3 | 10 | 3 | 10 | 3 | 36 |
| Skewnees | −0.17 | 1.98 | 0.34 | 1.47 | 0.06 | 1.76 | −0.18 | 1.48 | 0.08 | 1.15 | −0.20 |
| Std.error Skewnees | 0.22 | 0.22 | 0.22 | 0.22 | 0.22 | 0.22 | 0.22 | 0.22 | 0.22 | 0.22 | 0.22 |
| Kurtosis | 0.05 | 4.81 | 0.26 | 2.54 | −0.52 | 3.54 | −0.67 | 1.92 | −0.64 | 1.05 | −0.80 |
| Std.error Kurtosis | 0.4 | 0.44 | 0.44 | 0.44 | 0.44 | 0.44 | 0.44 | 0.44 | 0.44 | 0.44 | 0.44 |

**Note:**
TMS List1, total words remembered from the first list; TMS List2, total words remembered from the second list; TMS List3, total words remembered from the third list; TMS List4, total words remembered from the fourth list; TMS List5, total words remembered from the fifth list; TMS ListN interferences, equal for each list and corresponds to the total number of interference words that were repeated by participants but not present in the TMS word lists.

following cutoff criteria: SRMR < 0.08; RMSEA < 0.08; and CFI > 90; and TLI > 0.95 (*Hu & Bentler, 1999*).

Given the limitations to the use of the Cronbach's α (*McNeish, 2018*; *Revelle & Zinbarg, 2009*; *Sijtsma, 2009*; *Trizano-Hermosilla & Alvarado, 2016*), due to the very restrictive assumptions it relies on, we assessed internal consistency through the McDonald' ω total (*Revelle & Zinbarg, 2009*; *Zwick & McDonald, 2000*).

Finally, to test construct validity, we built a latent variable model with four latent variables: the two TMS latent variables (EF_TMS and M_TMS) described in the Hypothesized Model and two latent variables for the Classical neuropsychological tests, one reflected by the Executive Functions measures (EF_CNT) and one reflected by the memory measures (M_CNT). We then let correlate the four latent variables and tested whether the latent variables of similar constructs (*e.g.*, EF_TMS and EF_CNT) had a higher correlation coefficient as compared to the latent variables of distinct constructs (*e.g.*, EF_TMS and M_CNT). We tested this hypothesis through a unidirectional test on the z test on the standardized coefficients.

The authors have permission to use this tool.

## Data availability statement

Data, R script for the analyses conducted in R, and the .omv file for the analyses conducted in Jamovi are available on the OSF platform: https://osf.io/p4ruj.

## RESULTS

Descriptive analysis about lists of TMS tests shows how the mean number of words repeated by subjects increased from TMS-1 to TMS-4 and how it decreased from TMS-4 to TMS-5, as shown in Table 1. The average amount of words that subjects repeated but that

**Table 2  Descriptive statistics of neuropsychological tests.**

|  | MMSE raw | WMS Imm tot | WMS Delay tot | Stroop C W | MWSC | VFtot | SFtot | Digit Span F | Digit Span B |
|---|---|---|---|---|---|---|---|---|---|
| N | 121 | 121 | 121 | 121 | 121 | 121 | 121 | 121 | 121 |
| Mean | 29.10 | 25.20 | 22.40 | 36.20 | 4.99 | 35.09 | 49.50 | 4.66 | 3.49 |
| Median | 30 | 23.0 | 19.0 | 37 | 6 | 36 | 48.0 | 5 | 3 |
| Standard deviation | 1.59 | 13.30 | 16.10 | 14.10 | 1.68 | 12.10 | 14.9 | 1.59 | 1.18 |
| Skewness | −2.00 | 0.52 | 0.70 | −0.20 | −1.65 | −0.18 | 1.98 | −0.8 | 0.06 |
| Std. error Skewness | 0.22 | 0.22 | 0.22 | 0.22 | 0.22 | 0.22 | 0.22 | 0.22 | 0.22 |
| Kurtosis | 3.24 | −0.98 | −0.12 | 0.16 | 1.54 | −0.42 | 11.6 | −0.52 | −1.27 |
| Std. error Kurtosis | 0.44 | 0.44 | 0.44 | 0.44 | 0.44 | 0.44 | 0.44 | 0.44 | 0.44 |

Note:
MMSE raw, raw score of mini mental state examination; WMS imm tot and delay tot, Wechsler memory scale immediately and delay; Stroop C W, stroop colow word test; MWSC, modified Wisconsin sorting card; VF tot, verbal fluency total score; SF tot, semantic fluency total score; Digit Span F and B, digit span forward and backward.

**Table 3  Correlation matrix TMS lists with age and education.**

|  |  | Age | Education | TMS List 1 tot | TMS List 2 tot | TMS List 3 tot | TMS List 4 Tot | TMS List 5 tot | TMS tot list |
|---|---|---|---|---|---|---|---|---|---|
| Age | Pearson's r | — |  |  |  |  |  |  |  |
|  | p-value | — |  |  |  |  |  |  |  |
| Education | Pearson's r | −0.47*** | — |  |  |  |  |  |  |
| TMS List 1 tot | Pearson's r | −0.67*** | 0.49*** | — |  |  |  |  |  |
| TMS List 2 tot | Pearson's r | −0.52*** | 0.57*** | 0.59*** | — |  |  |  |  |
| TMS List 3 tot | Pearson's r | −0.53*** | 0.36*** | 0.49*** | 0.41*** | — |  |  |  |
| TMS List 4 Tot | Pearson's r | −0.68*** | 0.53*** | 0.54*** | 0.54*** | 0.75*** | — |  |  |
| TMS List 5 tot | Pearson's r | −0.52*** | 0.39*** | 0.40*** | 0.44*** | 0.62*** | 0.70*** | — |  |
| TMS tot Lists | Pearson's r | −0.72*** | 0.57*** | 0.72*** | 0.71*** | 0.84*** | 0.90*** | 0.81*** | — |

Notes:
*** $p < 0.001$.
Age, age of participants; Education, level of education of participants; TMS List 1 tot, total score of list 1 of TMS; TMS List 2 tot, total score of list 2 of TMS; TMS List 3 tot, total score of list 3 of TMS; TMS List 4 tot, total score of list 4 of TMS; TMS List 5 tot, total score of list 5 of TMS; TMS tot List, total score of all lists of TMS.

did not belong to any TMS word list, called "Interference" words, was also measured (Interference List 1 = 0.298 ± 0.542; Interference List 2 = 0.388 ± 0.583; Interference List 3 = 0.339 ± 0.571; Interference List 4 = 0.504 ± 0.732; Interference List 5 = 0.645 ± 0.773). Table 2 shows descriptive analysis about neuropsychological tests, in detail the mean and standard deviation of the subjects' scores on each neuropsychological test.

A significant correlation was found between the total score of each of the word lists (TMS tot List), the score of each list, and age and education level (Table 3). As shown in Table 4, from the comparison between models, our hypothesized bi-dimensional is the one that outperforms tridimensional and the unidimensional model in terms of fit indices. Furthermore, when comparing the hypothesized model to the alternative models using the Bayesian Information Criterion BIC (*Schwarz, 1978*) the hypothesized model showed the lowest BIC (2173.484 *vs.* ≥ 2178.213). Furthermore, as shown in Table 4, the hypothesized model had excellent fit indexes. In Fig. 1, we show the path diagram of the assumed models (Panel A, B, C, D, and E). In Table 5, we show the factor loading about our hypothesized two factors model.

**Table 4 Comparisons between models.**

|  | AIC | BIC | TFI robust | CFI robust | RMSEA robust | SRMR |
|---|---|---|---|---|---|---|
| Hypothesized Two-factor model | 2,142.73 | 2,173.48 | 1 | 1 | 0 | 0.017 |
| First alternative model (tri-dimensional) | 2,144.66 | 2,178.21 | 0.996 | 0.999 | 0.031 | 0.017 |
| Second alternative model (uni-dimensional) | 2,159.27 | 2,187.22 | 0.881 | 0.941 | 0.168 | 0.064 |
| Third alternative model (bi-dimensional) | 2,160.78 | 2,191.54 | 0.847 | 0.939 | 0.191 | 0.062 |
| Fourth alternative model (bi-dimensional) | 2,159.27 | 2,187.22 | 0.881 | 0.941 | 0.168 | 0.064 |

Note:
AIC, Akaike information criterion; BIC, Bayesian information criterion; TFI robust; CFI robust, comparative fit index; RMSEA robust, root mean square error of approximation; SRMR, standardized root mean square.

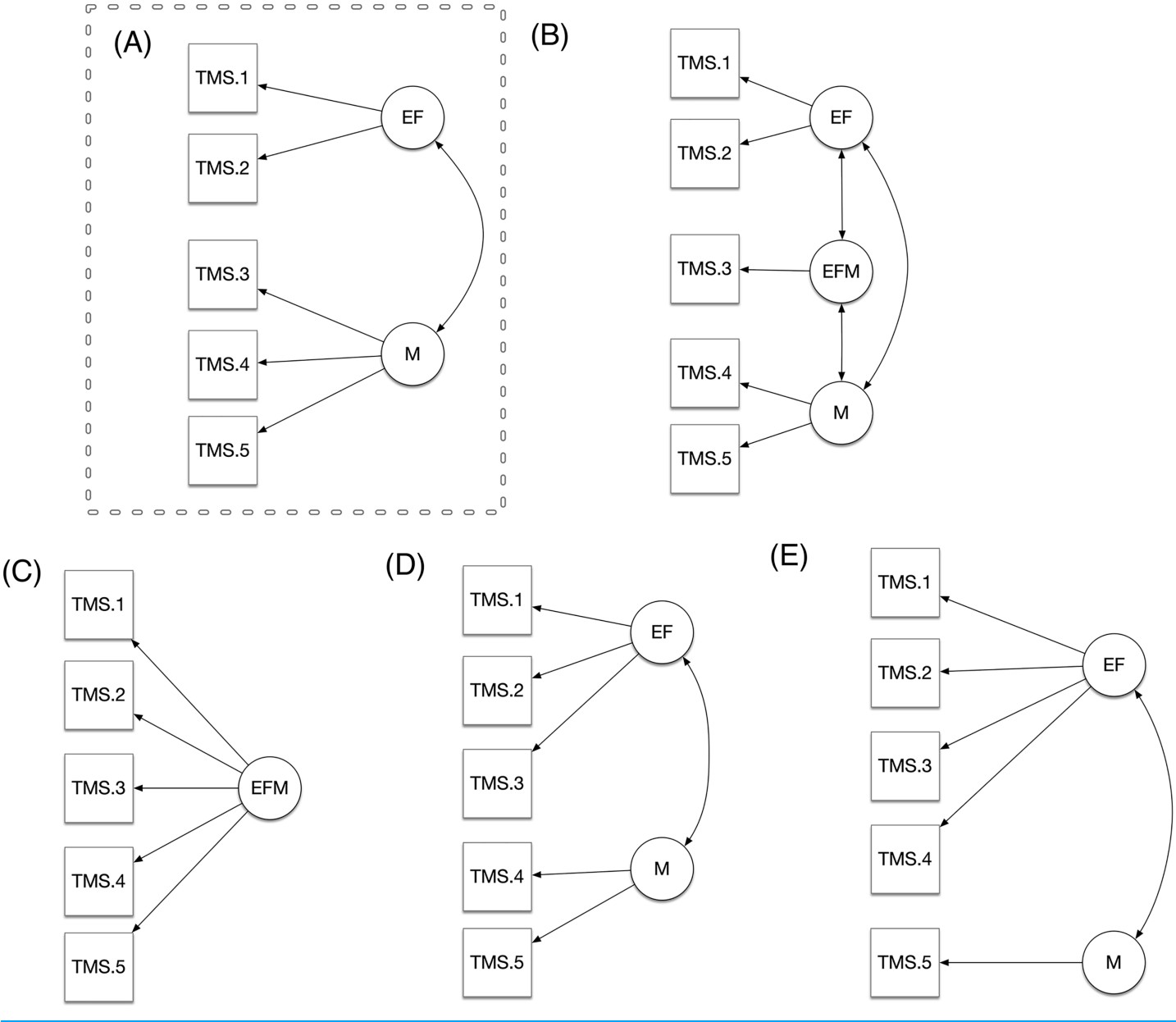

**Figure 1 (A–E) Comparison of models.**

**Table 5 Factor loading. Hypothesized two factors model (A).**

| Factor | Indicator | Estimate | SE | Z | p | Stand. estimate |
|--------|-----------|----------|------|-------|--------|-----------------|
| EF | TMS List 1 | 1.11 | 0.127 | 8.69 | <0.001 | 0.780 |
| | TMS List 2 | 1.14 | 0.135 | 8.45 | <0.001 | 0.759 |
| M | TMS List 3 | 1.56 | 0.153 | 10.19 | <0.001 | 0.802 |
| | TMS List 4 | 2.04 | 0.160 | 12.74 | <0.001 | 0.933 |
| | TMS List 5 | 1.50 | 0.159 | 9.40 | <0.001 | 0.756 |

Note:
EF, executive functions; M, memory; TMS List 1, total sum of words from the first list remembered; TMS List 2, total sum of words from the second list remembered; TMS List 3, total sum of words from the third list remembered; TMS List 4, total sum of words from the fourth list remembered; TMS List 5, total sum of words from the fifth list remembered.

**Table 6 Correlation matrix between M and EF and classical neuropsychological tests.**

| | | EF | M |
|---|---|-----|-----|
| EF | Pearson's r | – | – |
| M | Pearson's r | 0.60*** | – |
| Att-Cal | Pearson's r | 0.45*** | 0.31*** |
| Recording | Pearson's r | −0.07 | −0.02 |
| Recall | Pearson's r | 0.12 | 0.2* |
| Stroop CW | Pearson's r | 0.53*** | 0.35*** |
| MWSC | Pearson's r | 0.53*** | 0.45*** |
| VF tot | Pearson's r | 0.69*** | 0.5*** |
| SF tot | Pearson's r | 0.56*** | 0.44*** |
| DSB | Pearson's r | 0.58*** | 0.57*** |
| DSF | Pearson's r | 0.62*** | 0.62*** |
| WMS Imm tot | Pearson's r | 0.68*** | 0.69*** |
| WMS Delay tot | Pearson's r | 0.64*** | 0.71*** |

Notes:
* $p < 0.05$.
*** $p < 0.001$.
EF, executive functions; M, memory; Att-Cal, attention and calculation; MWSC, modified Wisconsin sorting card; VF tot, verbal fluency total; SF tot, semantic fluency total; DSB, digit span backward; DSF, digit span forward; WMS Imm tot, Wechsler memory scale immediately total; WMS Delay tot, Wechsler memory scale delay total.

In terms of internal consistency, the EF subscale showed an acceptable reliability ($\omega_t = 0.74$), whereas the M subscale showed a good reliability ($\omega_t = 0.87$).

We found that both M and EF subscales were correlated with the scores obtained by the participants in each specific test for memory and executive functions (Table 6).

The ANOVA on the EF and M subscales as dependent variables for age groups showed that there was significant difference $F$ (28.5) $p = < 0.001$ among groups for EF, and that there was significant difference $F$ (29.1) $p = < 0.001$ among groups for M. Regarding the post-hoc results, Table 7 showed that only in the comparison between the participants aged between 18–30 (G1) and 31–50 (G2) there was no significance for the EF subscales, so also with regard to the *post hoc* relative to the M subscales, not there was significance in the group of participants between the ages of G1 and G2. Finally, we found that the covariation between the latent variable reflecting EF in the TMS and the latent variable reflecting EF in

**Table 7** *Post Hoc* comparison—age groups.

| Age groups | | Age groups | Mean difference | SE | df | Tvalue | PTukey |
|---|---|---|---|---|---|---|---|
| G1 | – | G2 | 0.182 | 0.423 | 117 | 0.429 | 0.973 |
| | – | G3 | 1.145 | 0.439 | 117 | 2.606* | 0.050 |
| | – | G4 | 2.682 | 0.427 | 117 | 6.279*** | <0.001 |
| G2 | – | G3 | 0.963 | 0.446 | 117 | 2.161 | 0.140 |
| | – | G4 | 2.500 | 0.434 | 117 | 5.766*** | <0.001 |
| G3 | – | G4 | 1.537 | 0.449 | 117 | 3.422* | 0.005 |

Notes:
* $p < 0.05$.
*** $p < 0.001$.
SE, standard error; df, degree of freedom; t test t. G1, Group of participants with ages between 18–30; G2, age from 31 to 50; G3, age from 51 to 60; G4, age from 61 to 89.

the Classical neuropsychological tests (0.94) was higher than the covariation between the latent variable reflecting EF in the TMS and the latent variable reflecting M in the Classical neuropsychological tests (0.84, diff = 0.10, $z = 1.69$, $p = 0.045$). Furthermore, the covariation between the latent variable reflecting M in the TMS and the latent variable reflecting M in the Classical neuropsychological tests (0.81) was higher than the covariation between the latent variable reflecting EF in the TMS and the latent variable reflecting M in the Classical neuropsychological tests (0.66, diff = 0.15, $z = 2.59$, $p = 0.005$).

# DISCUSSION

This study examined the validity and reliability of the TMS among Italian healthy adults. Results of this study suggested that the different performance throughout the progressive external organization of the TMS word lists, as expected, we found a general increase of the total remembered words from TMS-1 to TMS-4. This is an indication that the progressive external organization of the material alleviates from memory strategies and improves memory performance, as demonstrated as well in the Spanish and Portuguese populations (*Yubero et al., 2011*; *Fernandes et al., 2018*).

There was a tendency of a drop in performance between TMS-4 and 5. There is a well-known phenomenon in psychological task performance called the "practice effect" (*Donovan & Radosevich, 1999*; *French et al., 2016*; *Goldstein et al., 2007*). The more the subject repeats a task, the better is the performance on that task. Because in the last condition, the participants were informed about the existence of two categories, there were different instructions to the task, the participants could think that the last condition (TMS-5) is different from the previous ones, losing both the practice effect and the focus on task since the instructions are different. Indeed, some participants say they focused on understanding what the categories were. This hypothesis arises from the "clinical" observation made during the administration of the test.

Another possibility is that the categories used in TMS-5 were much harder to memorize than those in TMS-4, but this is unlikely, as words in all these categories had a similar frequency of use; or another possibility is the possible interference of words between the lists. Indeed, the number of intrusions increases as you get closer to List 5.

The hypothesis of identifying sample heterogeneity as one of the causes could also be considered. For all age groups, the score of the words remembered from the TMS 1 list to the TMS 4 list increases, as already highlighted. From the TMS 4 list to the TMS 5 list, the number of words remembered decreases for all groups but especially for the older ones. These results are consistent with the ones found in *Yubero et al. (2011)*, in which none of the groups showed statistically significant differences between TMS-4 and TMS-5, although there was a reduction in the number of words recalled in TMS-5. Instead, *Fernandes et al. (2018)* find a general increase in the total words remembered from TMS-1 to TMS-5. Like those of Fernandes, our results seem to be consistent with those of *Bor et al. (2003)*, who demonstrated that coding strategies improve memory performance in complex tasks.

Concerning the structure of the test, the results of CFA suggested that the two-factor model is an acceptable one, with TMS list 1 and TMS list 2 reflecting one factor, and TMS list 3, TMS list 4, and TMS list 5 reflecting another factor. In line with *Yubero et al. (2011)* and *Fernandes et al. (2018)*, our results support the idea of the TMS as a measure of memory and executive functions. This may indicate that the use of the total score could be appropriate, calculating both M and EF score simultaneously. Consequently, the TMS appears to be particularly suitable for research and clinical purposes because it offers an efficient way to measure general EF and M functions. Concerning the psychometric properties, the Italian TMS showed internal consistency values that ranged from good to excellent and in line with those reported for the original TMS (*Fernandes et al., 2018*; *Yubero et al., 2011*).

Our analyses showed that the latent variable reflected by the EF TMS items correlated more with the latent variable reflected by the EF Classical neuropsychological tests items than with the Memory Classical neuropsychological tests. *Vice versa*, the latent variable reflected by the Memory TMS items correlated more with the latent variable reflected by the Memory Classical neuropsychological tests items than with the EF Classical neuropsychological tests. Such a pattern of results supports the construct validity in terms of convergent and discriminant validity.

*Yubero et al. (2011)* found that the level of education in the age grouping influences the score obtained on the TMS and the elderly population has difficulty performing tasks where a self-initiation process is required for efficient coding and information retrieval. In our case, an analysis was not carried out by dividing by age groups because the sample size did not allow it.

As indicated before, classical test of episodic memory does not consider the separate assessment of memory and executive functions and therefore they are deeply influenced by the patient's ability to mobilize complex memory strategies. The TMS is trying to avoid this source of confusion to interpret the results of an episodic memory test by progressively reducing the need of executive functions. Other memory test as the California Verbal Learning test is evaluating the use of memory strategies as well. However, the CVLT, is assessing the use of memory strategies by the subjects' answers, but not by the manipulation of the verbal material itself. The TMS is progressively organizing the verbal material to reduce at minimum the need of EF and evaluate episodic memory without the

influence of internal cognitive strategies. As an example, in the last condition of the TMS two categories are presented one after the other and the participant is aware about that external organization of the material to avoid the use of any internal memory strategy. In this sense, the material organization performed in the TMS directly evaluates the memory abilities expecting having worse performance in patients with a primary memory deficit.

Some shortcomings of the present study must be mentioned. First, the sample is small, it is not strongly representative of the entire Italian population, and the distribution of the sample is too homogeneous. Indeed, one of our future objectives is to expand the sample and make it representative of the healthy Italian and, subsequently, also add a clinical population. Additionally, the findings regarding the CFA must be interpreted with caution, and future studies on measurement invariance to the clinical *vs.* non-clinical population are therefore needed (see *Yubero et al., 2011* for a clinical study). Considering the discrepancy between the models developed by us and the one proposed by *Fernandes et al. (2018)*, In the future it would be useful to carry out an administration to a large Spanish and Italian sample in order to analyze both and detect the cause of this differentiation.

In conclusion, our preliminary study shows that the Italian version of TMS is a relatively valid and reliable measure of executive and memory functions in its Italian version. TMS is useful for improving cognitive diagnosis in patients with neurodegenerative diseases when a memory failure can be due to both executive dysfunction and a primary memory problem. This test is useful for detecting slight differences in cognitive functioning, especially because other neuropsychological instruments do not test the relationships between memory and executive functions. The profiles from the TMS can be easily used for the development of a personalized or even more tailored rehabilitation program.

The TMS could be useful in prospective studies in children and school assessments to understand if a learning deficit is memory or executive function related and, therefore, help structure the school educational-rehabilitation plans. TMS could have excellent practical implications in the clinical field. For example, it could be very useful in those neurodegenerative diseases in which it is still difficult to make a differential diagnosis, such as the subtypes of progressive nuclear paralysis. In these subtypes of PSP and diseases for example, there is still a lot of confusion about the characteristic cognitive deficits. The greatest confusion is precisely between deficit to executive functions and memory deficits, for example (*Vaccaro et al., 2020*). Typically, low scores in an episodic memory test are used for starting an episodic memory intervention. However, this intervention could fail if the primary deficit is an executive function deficit rather than memory. Therefore, TMS could help clinicians to find a more appropriate intervention approach improving the rate of success in neuropsychological rehabilitation.

Therefore, it might be used in future studies where it is crucial to either measure or control for the joint and differential contribution of memory and executive functions (*El Haj et al., 2016*; *Gray et al., 2012*; *Novellino et al., 2019*; *Polak et al., 2012*; *Vaccaro et al., 2018*; *Sarica et al., 2021*; *Vaccaro et al., 2020*).

### Funding

The authors received no funding for this work.

### Competing Interests

The authors declare that Prof. Marco Tullio Liuzza is an Academic Editor for PeerJ.

### Author Contributions

- Maria Grazia Vaccaro conceived and designed the experiments, performed the experiments, analyzed the data, prepared figures and/or tables, authored or reviewed drafts of the article, cognitive test administration, and approved the final draft.
- Marco Tullio Liuzza conceived and designed the experiments, analyzed the data, prepared figures and/or tables, authored or reviewed drafts of the article, and approved the final draft.
- Massimiliano Pastore analyzed the data, authored or reviewed drafts of the article, and approved the final draft.
- Nuria Paúl conceived and designed the experiments, authored or reviewed drafts of the article, and approved the final draft.
- Raquel Yubero conceived and designed the experiments, authored or reviewed drafts of the article, and approved the final draft.
- Andrea Quattrone performed the experiments, authored or reviewed drafts of the article, clinical selection of the control group and materials, and approved the final draft.
- Gabriella Antonucci analyzed the data, authored or reviewed drafts of the article, and approved the final draft.
- Antonio Gambardella analyzed the data, authored or reviewed drafts of the article, and approved the final draft.
- Fernando Maestú conceived and designed the experiments, authored or reviewed drafts of the article, and approved the final draft.

### Human Ethics

The following information was supplied relating to ethical approvals (*i.e.*, approving body and any reference numbers):

CEIC Hospital Clínico San Carlos (Madrid – Spain) with number C.I. 18/422-E_BS.

### Data Availability

The data is available at the Open Science Framework: Liuzza, Marco Tullio, and MARIA G VACCARO. 2022. "Preliminary Contribution to the Validation of the Italian Version of the Test of Memory Strategies." OSF. September 2. osf.io/p4ruj.

### Supplemental Information

Supplemental information for this article can be found online at http://dx.doi.org/10.7717/peerj.14059#supplemental-information.

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
