# Peer review of "The validity and reliability of the Test of Memory Strategies among Italian healthy adults"

_PeerJ, doi:10.7717/peerj.14059_

## Round 0.1 · original submission · Major Revisions

Please revise the paper carefully following reviewers' suggestions.

·

Basic reporting

The article is generally well written and well-formatted.
Tables are not always correctly formatted

Experimental design

The research question should be directly related to the present study. The last paragraph of the introduction, l.125-135, focused on "memory strategies in old adults" whereas the present study did not target older adults specifically. The lack of Italian adaptation is not a well-appropriate argument to justify the missing data on this topic. In my view, the adaption in Italian is self-sufficient.

More details should be added to the procedure, for instance, is there a delay between the presentation of the lists?

Validity of the findings

The validity of the findings is limited by the heterogeneity of the sample (see below).
A correction should be applied for multiple testing (correlations).
The discussion should be more qualified (see below).

Additional comments

### Major comments
The article entitled "Preliminary contribution to the adaptation of the Italian version of the Test of Memory Strategies" presents the Italian adaptation and validation of a memory test (TMS) focusing on the interplay between episodic memory and executive functions. There are very few memory tests that offer to control how executive functions, here memory strategies, contribute to episodic memory. Yet, these two functions can be conjointly or independently impaired. Therefore, the present contribution is relevant and useful for clinicians and researchers in the field. Overall, the adaptation is well conducted and the type of analyses sounds appropriate, but there is a lack of precision for the presentation of the memory test and of the sample and the statistical analyses and the results should be revised. In its present form, I would not recommend the publication of the article for the following reasons (by order of importance).

1. The sample appears very heterogenous with age ranging from 18 to 89 years and education being around 13 years +/- 4 years. There is no clear presentation of the sample about sex ratio, mean (SD) for the MMSE, distribution of age and education (maybe add histograms), and so on. And more importantly, these variables, especially age and education, are not considered in the statistical analyses. Such a disparity precludes any conclusion to be drawn as aging is known to alter memory and education moderating executive functioning. The authors stated l.333 "In our case, an analysis was not carried out by dividing by age groups because the sample size did not allow it". Even if a stratification strategy is not possible with their sample, it still remain possible to conduct correlation (and even regression analysis) with age and education (as continuous variables) on the one hand and with the different memory scores, on the other hand. It is mandatory to have more information about the distribution of these variables within the sample and knowing the possible relationships between these variables and the TMS.
2. The conclusion drawn from the convergent validity test should be more precautionary discussed (see l.319 and follow). Indeed, the two components of the TMS (memory and executive functions) are related to almost the same classical executive and episodic memory tests, without any specificity. Even if each component is "more strongly related" (see l.321) with some other scores, no statistical tests are reported to support such a conclusion. In addition, no correction seems to be applied for multiple comparisons which may lead to false positive results given the number of correlation tests realized. I recommend conducting the correlation analysis with such a correction and to qualify the discussion about the convergent validity.
3. The introduction would gain in clarity by presenting, explicitly, the TMS and how executive functions are manipulated within the test. In the same vein, more precision about the procedure would be welcomed, for instance, is there a delay between the presentation of the lists.
4. The discussion may have a greater reach by discussing how the manipulation of executive functions in a parametric manner is helpful compared to more classical tests such as the California Verbal Learning Test.
5. The discussion about the "tendency of a drop in performance between TMS-4 and 5" is too speculative. It seems rather complicated to imagine that the participants are "losing both the practice effect and the focus on task". Have the authors considered simpler explanation such as fatigue or discrepancies due to age heterogeneity? Knowing how the data are distributed (e.g. boxplots) may provide some insights (different subgroups of respondents).

### Minor comments
- l.56: please specify which kind of memory is concerned, at least for the first mention (i.e. episodic memory)
- l.59: "It is well known that the circuits supporting M functions and EF are well interconnected", the circuits and main brain regions involved in episodic memory and executive functions should then be presented in the introduction as well.
- l.62-63: the sentence let’s suppose that executive functions on solely (or at least mainly) involved in memorization (i.e. encoding), but numerous studies have reported their importance also at retrieval (e.g. in aging Craik et al., 2018). The sentence should be updated accordingly.
- l.72-73: There is no clear transition between the two ideas/paragraphs.

## References
Craik, F. I., Eftekhari, E., Bialystok, E., & Anderson, N. D. (2018). Individual differences in executive functions and retrieval efficacy in older adults. _Psychology and Aging_, _33_(8), 1105.

·

Basic reporting

Basic reporting contains minor language mistakes.

Experimental design

Method and results sections needs improvements.

Validity of the findings

Findings are valid.

Additional comments

Thanks for opportunity to review manuscript entitled ‘‘Preliminary contribution to the adaptation of the Italian version of the Test of Memory Strategies’’ for Peerj Journal. The authors translated Test of Memory Strategies to Italian cultural context and examined the validity and reliability of this test in a nonclinical sample of Italian adults. According to my Google citation analyses of this instrument, the strength of the manuscript includes examining factor structure of this scale using confirmatory factor analyses and expanding psychometric studies regarding to this scale with an article written in English language as well as Italian cultural context. Overall, although the article is generally well-written and deserves to be published in this journal, some necessary and minor revisions required before publication of this article. Because my main philosophy of reviewing a manuscript as reviewer and sometimes an editor to improve the manuscript and not punishing the authors, I provided very specific and detailed peer review of the manuscript to increase its quality and citation potential. I hope authors of the manuscript may benefit from my review. Necessary revisions reported section by section with the page and line number and when possible with suggestions.
Necessary Revisions
Title
1. Page 6, Line 1-2: I think title of the article need to be revised for better reflect to study. Translation is a process conducted as a part of linguistic validity. Moreover, reliability analyses also conducted in this study. Thus, one revision may be that ‘ ‘The validity and reliability of Test of Memory Strategies among Italian healthy adults’’
Abstract
2. Page 7, Line 34: Sample is unclear in aim sentence. Thus, revising sample as ‘ ‘adult sample’’ in a better choice.
3. Page 7, Line 34: According to APA 7 rules, sentence must no begin with numbers. Thus, authors must correct 121 as One hundred twenty one
4. Page 7, Line 34: Reporting of mean and standard deviation is wrong according to APA 7 rules. Thus, following sentence must be revised as ‘‘One hundred twenty one healthy participants (Mage = 45 years old, SD = 20.4) who underwent a neuropsychological examination were involved in this study.’’ Moreover, it is almost impossible to be mean age exactly 45. Thus, authors also need to report to mean age and standard deviation with two decimal.
5. Page 7, Line 36: Authors also examined convergent validity of Test of Memory Strategies using Pearson correlation analyses and reliability using McDonald’s omega. However, they only give information about confirmatory factor analysis. Thus, following sentences need to be revised ‘ ‘Confirmatory Factor Analyses were employed to evaluate construct validity.’’ One revision may be that ‘ ‘Confirmatory factor analyses were employed to evaluate construct validity of TMS. Pearson correlation analyses were used to examine convergent validity of TMS scores. McDonald’s omega used to examine internal consistency.’’
6. Page 7, Line 37: Authors need to change assumption in this sentence with expectation as ‘ ‘….the assumption that the TMS-1 and TMS-2’’. New form looks like ‘ ‘….the expectation that the TMS-1 and TMS-2…’’. Assumption indicates different things in research design (non-testable) and statistics (testable). Thus, it must use with caution.
7. Page 7, Line 39: Authors need to change assumption in this sentence with predictions ‘ ‘….in line with the assumption that TMS-1 and TMS-2…..’’ New form looks like ‘ ‘….in line with the prediction that TMS-1 and TMS-2…..’’.
8. Page 7, Line 39: The following sentence is awkward and need revision. ‘ ‘ The findings suggest that TMS is an adequate measure to assess M and EF while simultaneously presenting preliminary adequate psychometric properties in an Italian sample.’’ One revision may be that ‘‘This preliminary findings suggest that TMS is a valid and reliable scale to simultaneously assess M and EF among Italian healthy adults.’’
9. Page 7, Line 49-50: I think authors must remove simultaneous investigation from keywords and must add Italian healthy adults after the psychometric validation
10. Apart from above correction, Abstract section is well written.
Introduction
11. Page 7, Line 59: I think authors must remove well from following sentence ‘ ‘……well interconnected’’
12. Page 7, Line 60-63: I think authors must remove following sentence from the manuscript as they are unrelated to content, difficult to understand and distort flow of paragraph ‘ ‘ From an anatomical point of view, this evidence indeed has a functional consequence, indicating the in-time collaboration between these two cognitive processes. EF and M are intertwined: EF plays an essential role in memorization strategies.’’
13. Page 8, Line 67-68: The citation needs for following sentence ‘ ‘A mediation analysis revealed that the EF network had an indirect positive effect on episodic memory performance in the amnestic mild cognitive impairment patients.’’
14. Page 8, Line 84-85: I think authors must remove following sentence from the manuscript as it mind-confusing. ‘ ‘- which includes measures of memory strategies, (Scarpa et al., 2006) -,’’.
15. Page 9, Line 109-110: It is unclear what authors want to mean with levels in following sentence ‘ ‘between different age groups and levels.’’. Do authors mean education levels?
16. Page 9, Line 112: Please correct ‘ ‘Fernandes et al. like Yubero et al.’’ as ‘ ‘Fernandes et al., like Yubero et al.,’’
17. Page 9, Line 122-123: I think authors must remove following from the sentence ‘ ‘such as Yubero et al. (Yubero et al., 2011)’’.
18. Page 9, Line 127: I think authors ‘‘ old adults’’ must be ‘‘older adults’’ in the sentence.
Method
19. Page 10, Line 142-143: The following sentence is awkward and need revision. ‘ ‘A sample of 121 participants was recruited between 18 and 89 years (mean age = 45 years old, SD = 20.4; years of schooling = 13.2±3.97).’’ Moreover, it is almost impossible to be mean age exactly 45. Thus, authors also need to report to mean age, years of schooling and standard deviation with two decimal. Moreover, please report like this (Mage = 45.xx, SD = 20.4x)
20. Page 10, Line 148: The citation needs for cut-off score of The Mini Mental State Examination. Is 24 also valid in Italian cultural context?
21. Page 10, Instruments section: Instrument section must rearrange with subtitles. In this form, it is very messy. Moreover, author must give information used standardized neuropsychological tests. Are they valid and reliable in Italian cultural context? Moreover, authors give at worst possible scores of these tests as well as meaning of higher scores. The same thing also valid for Test of Memory Strategies. Authors must add possible score of each list and meaning of higher scores.
22. Page 12, Line 227-229: Authors must divide following sentences to two separate sentences. Correlation coefficients are not a descriptive statistics. ‘ ‘Descriptive statistics of neuropsychological battery and TMS were calculated for the whole sample, including frequencies, means, and correlation matrix between the scores of each list and between other psychological tests.’’
23. Page 12, Line 230: ‘‘Confirmatory Factor Analysis’’ must be ‘ ‘confirmatory factor analysis’’.
24. Page 12, Line 231: Authors did not indicate hypotheses in Introduction section. Thus, following must remove from manuscript ‘ ‘and our hypothesis that’’.
25. Page 12, Statistical Analyses: Authors must explain each tested competing models and reasons behind testing it in Statistical analyses section. Without theory and empirical findings, testing alternative models are not defensible. Moreover, based on principal component analysis of Fernandes et al., authors must add a competing model that included first four list as EF and TMS-5 as M.
26. Page 12, Statistical Analyses: Which estimation method used in confirmatory factor analyses must be added to statistical analyses section. Moreover, authors must add brief information about assumption testing process in confirmatory factor analyses and correlation analyses.
27. Page 12, Line 239: the writing of MacDonald’ ⍵ total is wrong and must be corrected as McDonald’ omega (⍵)
28. Page 12, Line 239: What authors want to mean with fit index of CFA (Green & Yang, 2009) ? , it is not possible to test an instrument’s’ internal consistency with fit indexes. Authors must remove this wrong information.
29. Page 12, 241-247: Authors must move following information after the sentence of With a set of confirmatory factor analysis (CFA) we compared different models in order to identify the model that best explains our data and our hypothesis that the TMS subtests reflected two-dimensional structure with two latent variables, that we have named as EF, and M. ‘ ‘
We have considered the total score of each single 5 lists as the 5 definitive variables, and we have tested whether the hypothesized model where the EF factor was reflected by the first two TMS list scores, and the M factor was reflected by the TMS list scores number 3, 4 and 5 performed better than other three alternative models. The first model was a tridimensional model (Panel C) where a third factor reflected by the TMS 3 score was created. The second model was a unidimensional model (Panel D). Finally, we created an alternative two-dimensional model (Panel B) in which list was considered a variable reflecting the factor EF. ‘’
30. Page 12, 247: Authors must remove following sentence no need to write this. ‘ ‘Authors have permission to use this tool.’’
Results
31. Results, General: Authors inconsistently reported statistical findings along them manuscript and in the tables. Authors must report all study findings with two decimals along the manuscript except for p values which must be three decimals.
32. Results, Table 1: Table 1 is unnecessary and must remove from the manuscript and all related information. Authors examine psychometric properties of Test of Memory Strategies not other tests. Hus authors must remove following lines from the results ‘ ‘Table 1 reports descriptive statistics of neuropsychological battery (MMSE total; WMS immediate and delay total; Stroop test; MWSC test; VF test; SF test; Digit span forward and backward test for the global sample.’’ And from statistical analysis section following information ‘ ‘of neuropsychological battery and’’.
33. Results, Table 2. Authors must add minimum and maximum scores to Table 2. Moreover, title must be revised such that ‘ ‘Descriptive statististics about subscales and total scores of TMS’’. N must be italic in the Table 2. Mean must be M (italic) and standard deviation must be SD (italic). Notes must be under the table.
34. Results, Table 3, Table 4: The order of Table 3 and Table 4 must be changed. Moreover, authors first give information about goodness on fit indexes of competing models and then result of chi-square nested difference tests in Results section. Additionally, author must add information about using chi-square nested difference test to statistical analysis section. This information is completely missing in that section.
35. Page 12, 267: In this line authors indicated that ‘ ‘In Figure 1 we show the path diagram and the standardized coefficients of our hypothesized model.’’. However, there is no information about standardized factor loadings in the figure. Author must add narratively information about factor loadings for best fitting model and their significance levels to Results section. Moreover, author must add a supplementary material for all competing models standardized factor loadings as well latent factor correlations (if available) as well as item means standards deviations and inter-item correlations of TMS. Without this findings, it is not possible to demonstrate correctness of statistics as well as evaluate comprehensively to alternative models.
Discussion
37. Page 13, 278-285: Following sentences is not appropriate for Discussion section but rather may be used is Introduction section ‘ ‘Sometimes, in the adult and childhood clinical population, it is difficult to understand if there is a memory or executive function impairment or both just from the interpretation of traditional neuropsychological tests. This study aimed to improve and increase the neuropsychological assessment tools available in Italy to cognitive diagnosis using the TMS, a neuropsychological test already validated in Spanish and Portuguese (Yubero et al., 2011; Fernandes et al., 2018). There is no neuropsychological test for evaluating the interaction between memory skills and executive functions simultaneously in the Italian context. This preliminary study on the Italian population could fill the gap in the neuropsychological test resources for improving diagnosis.’’ Moreover, sometimes is not a word that may be used in a scientific article and must remove.
38. Page 13, Line 290-292: Following sentences is not appropriate for Discussion section and must remove. ‘ ‘Thus, in TMS 1 to 4, we can hypothesize that memory strategies are implicitly suggested by the subdivision in categories (externally facilitated), helping the subject in the encoding of the words.’’
39. Page 14, Line 304-306: Authors did not conduct an independent or dependent samples t-test but indicated that ‘ ‘These results are consistent with the ones found in Yubero et al. (2011), in which none of the groups showed statistically significant differences between TMS-4 and TMS-5, although there was a reduction in the number of words recalled in TMS-5.’’ How it can be possible?
40. Page 14, Line 319-320: Following sentence mut be revised as I understand from following sections authors only tested convergent validity of scale. However, author indicated in this sentence that they both used convergent and discriminant validity. ‘ ‘ Another aspect of construct validity — convergent/discriminant validity — was reflected by the correlations of the five TMS subtests with classical tests of the same/different constructs. ‘’ One revision may be that ‘ ‘Another aspect of construct validity -convergent validity — was demonstrated by the correlations of the five TMS subtest scores with neuropsychological tests measuring the similar constructs.’’
41. Page 14, Line 338-339: I am not able to understand what authors want to mean ‘ ‘Moreover, a functional study would be useful to study the correlation with neuroanatomy.’’
42. Page 14, Line 339, Page 15 339-341: The same sentence repeated two times and following must remove ‘ ‘An analysis of the interaction analysis by age group was not conducted in our work, considering the very small and too heterogeneous sample, unlike Fernandes' work.’’
43. Discussion, Limitation Section: limitations of study must significantly improve. Authors must add limitations related to sample (not representative Italian population), using self-report scales, cross-sectional research design, and large age range.
44. Discussion, General: Practical implication section is completely missing in the manuscript and must add after the implications.
45. Discussion, Page 15 350-362: Following sentences is not appropriate for Discussion section but rather may be used is Introduction section ‘ ‘Some tests investigate the executive functions and memory separately (such as Rey Auditory Verbal Test; etc.), but this does not allow to verify if during execution the subject fails due to a slowdown in memory or planning or problem-solving strategies. Instead, the TMS is structured to capture which phase of the test the subject increased the number of errors. A failure in the initial condition would but a clear improvement after diminished executive functions would signify an executive deficit more than a memory problem. If the participant does not improve across time, then a primary memory problem can be noticed.’’
46. All references are wrong as per Peerj rules.
47. Authors must rearrange Table 5 as below, in this form contains a lot of repetition.

EF M
r r
EF - -
M 0.60*** -
Att-Cal 0.45*** 0.31***




Notes. EF = executive functions; M = memory;………. *p <.05, ** p < .01, *** p < .001.

---

## Round 0.2 · Minor Revisions

I am writing to inform you that your manuscript - The validity and reliability of the Test of Memory Strategies among Italian healthy adults - needs minor revisions to be accepted for publiction. Please see reviewers’ comments.

·

Basic reporting

no comment

Experimental design

no comment

Validity of the findings

no comment

Additional comments

The authors have successfully addressed all my previous concerns.

·

Basic reporting

Clear and unambiguous, professional English used throughout.

Experimental design

Method and results section is correct.

Validity of the findings

The findings is valid.

Additional comments

Thanks for opportunity to rereview manuscript entitled ‘‘The validity and reliability of the Test of Memory Strategies among Italian healthy adults’’ for Peerj Journal. Authors significantly improved manuscript from the first review with a good will. Some minor revisions left behind before publication of article.
1. Bonferroni correction required to use all inferential statistics not some part of inferential statistics. Thus, researchers can check and correct this problem ıf they want to use it. More specifically, author must determine significance level based on all tests in Table 3, Table 6 and Table 7 not a specific test. As a multivariate analyst, I don’t recommend Bonferroni correction in a sample size like this. If required to use, Benjamini-Hochberg false discovery rate with a .05 error rate may be used. However, it will very complicate the significance levels. I recommend to present findings without corrections. This comment also related to editor. Authors may make their decisions after consulting the article editor.
2. Authors must upload latest version of Tables after the corrections.
3. After to Bonferroni correction, is significance level still correct in Table 3, Table 6? authors must check it carefully.
4. Add following under to Table 7 *p<.05, ** p< .01, ***p<.001. Moreover, add asterisks for 0.050, <.001, 0.005, ıf significant after Bonferroni correction.
5. Add an introduction sentence to Discussion section first paragraph first sentence such that This study examined the validity and reliability of the TMS among Italian healthy adults. Results of this study suggested that…..

---

## Round 0.3 · accepted · Accept

I am writing to inform you that your manuscript - The validity and reliability of the Test of Memory Strategies among Italian healthy adults - has been Accepted for publication. Congratulations!